# ROBUST LOCAL FEATURES FOR IMPROVING THE GENERALIZATION OF ADVERSARIAL TRAINING

**Chuanbiao Song & Kun He** *& **Jiadong Lin**
School of Computer Science and Technology
Huazhong University of Science and Technology
Wuhan, 430074, China
{cbsong,brooklet60,jdlin}@hust.edu.cn

**Liwei Wang**
School of Electronics Engineering
and Computer Sciences, Peking University
Peking, China
wanglw@cis.pku.edu.cn

**John E. Hopcroft**
Department of Computer Science
Cornell University, NY 14853, USA
jeh@cs.cornell.edu

## ABSTRACT

Adversarial training has been demonstrated as one of the most effective methods for training robust models to defend against adversarial examples. However, adversarially trained models often lack adversarially robust generalization on unseen testing data. Recent works show that adversarially trained models are more biased towards global structure features. Instead, in this work, we would like to investigate the relationship between the generalization of adversarial training and the robust local features, as the robust local features generalize well for unseen shape variation. To learn the robust local features, we develop a Random Block Shuffle (RBS) transformation to break up the global structure features on normal adversarial examples. We continue to propose a new approach called Robust Local Features for Adversarial Training (RLFAT), which first learns the robust local features by adversarial training on the RBS-transformed adversarial examples, and then transfers the robust local features into the training of normal adversarial examples. To demonstrate the generality of our argument, we implement RLFAT in currently state-of-the-art adversarial training frameworks. Extensive experiments on STL-10, CIFAR-10 and CIFAR-100 show that RLFAT significantly improves both the adversarially robust generalization and the standard generalization of adversarial training. Additionally, we demonstrate that our models capture more local features of the object on the images, aligning better with human perception.

## 1 INTRODUCTION

Deep learning has achieved a remarkable performance breakthrough on various challenging benchmarks in machine learning fields, such as image classification (Krizhevsky et al., 2012) and speech recognition (Hinton et al., 2012). However, recent studies (Szegedy et al., 2014; Goodfellow et al., 2015) have revealed that deep neural network models are strikingly susceptible to *adversarial examples*, in which small perturbations around the input are sufficient to mislead the predictions of the target model. Moreover, such perturbations are almost imperceptible to humans and often transfer across diverse models to achieve black-box attacks (Papernot et al., 2017; Liu et al., 2017; Wang et al., 2019; Lin et al., 2020).

Though the emergence of adversarial examples has received significant attention and led to various defend approaches for developing robust models (Madry et al., 2018; Dhillon et al., 2018; Wang & Yu, 2019; Song et al., 2019; Zhang et al., 2019a), many proposed defense methods provide few benefits for the true robustness but mask the gradients on which most attacks rely (Carlini & Wagner, 2017a; Athalye et al., 2018; Uesato et al., 2018; Li et al., 2019). Currently, one of the best

---
*Corresponding author

techniques to defend against adversarial attacks (Athalye et al., 2018; Li et al., 2019) is *adversarial training* (Madry et al., 2018; Zhang et al., 2019a), which improves the adversarial robustness by injecting adversarial examples into the training data.

Among substantial works of adversarial training, there still remains a big robust generalization gap between the training data and the testing data (Schmidt et al., 2018; Zhang et al., 2019b; Ding et al., 2019; Zhai et al., 2019). The robustness of adversarial training fails to generalize on unseen testing data. Recent works (Geirhos et al., 2019; Zhang & Zhu, 2019) further show that adversarially trained models capture more on global structure features but normally trained models are more biased towards local features. In intuition, global structure features tend to be robust against adversarial perturbations but hard to generalize for unseen shape variations, instead, local features generalize well for unseen shape variations but are hard to generalize on adversarial perturbation. It naturally raises an intriguing question for adversarial training:

*For adversarial training, is it possible to learn the robust local features , which have better adversarially robust generalization and better standard generalization?*

To address this question, we investigate the relationship between the generalization of adversarial training and the robust local features, and advocate for learning robust local features for adversarial training. Our main contributions are as follows:

- To our knowledge, this is the first work that sheds light on the relationship between adversarial training and robust local features. Specifically, we develop a Random Block Shuffle (RBS) transformation to study such relationship by breaking up the global structure features on normal adversarial examples.

- We propose a novel method called Robust Local Features for Adversarial Training (RLFAT), which first learns the robust local features, and then transfers the information of robust local features into the training on normal adversarial examples.

- To demonstrate the generality of our argument, we implement RLFAT in two currently state-of-the-art adversarial training frameworks, PGD Adversarial Training (PGDAT) (Madry et al., 2018) and TRADES (Zhang et al., 2019a). Empirical results show consistent and substantial improvements for both adversarial robustness and standard accuracy on several standard datasets. Moreover, the salience maps of our models on images tend to align better with human perception.

## 2 PRELIMINARIES

In this section, we introduce some notations and provide a brief description on current advanced methods for adversarial attacks and adversarial training.

### 2.1 NOTATION

Let $F(x)$ be a probabilistic classifier based on a neural network with the logits function $f(x)$ and the probability distribution $p_F(\cdot|x)$. Let $\mathcal{L}(F; x, y)$ be the cross entropy loss for image classification. The goal of the adversaries is to find an adversarial example $x' \in \mathcal{B}_\epsilon^p(x) := \{x' : \|x' - x\|_p \leq \epsilon\}$ in the $\ell_p$ norm bounded perturbations, where $\epsilon$ denotes the magnitude of the perturbations. In this paper, we focus on $p = \infty$ to align with previous works.

### 2.2 ADVERSARIAL ATTACKS

**Projected Gradient Descent.** Projected Gradient Descent (PGD) (Madry et al., 2018) is a stronger iterative variant of Fast Gradient Sign Method (FGSM) (Goodfellow et al., 2015), which iteratively solves the optimization problem $\max_{x':\|x'-x\|_\infty < \epsilon} \mathcal{L}(F; x', y)$ with a step size $\alpha$:

$$x^0 \sim \mathcal{U}\left(\mathcal{B}_\epsilon^\infty(x)\right),$$
$$x^{t+1} = \Pi_{\mathcal{B}_\epsilon^\infty(x)}\left(x^t - \alpha \operatorname{sign}\left(\nabla_x \mathcal{L}(F; x, y)|_{x^t}\right)\right), \tag{1}$$

where $\mathcal{U}$ denotes the uniform distribution, and $\Pi_{\mathcal{B}_\epsilon^\infty(x)}$ indicates the projection of the set $\mathcal{B}_\epsilon^\infty(x)$.

**Carlini-Wagner attack.** Carlini-Wagner attack (CW) (2017b) is a sophisticated method to directly solve for the adversarial example $x^{adv}$ by using an auxiliary variable $w$:

$$x^{adv} = 0.5 \cdot (\tanh(w) + 1). \tag{2}$$

The objective function to optimize the auxiliary variable $w$ is defined as:

$$\min_w \left\| x^{adv} - x \right\| + c \cdot \mathcal{F}\left(x^{adv}\right), \tag{3}$$

where $\mathcal{F}(x^{adv}) = \max\left(f_{y^{\text{true}}}(x^{adv}) - \max\left\{f_i(x^{adv}) : i \neq y^{\text{true}}\right\}, -k\right)$. The constant $k$ controls the confidence gap between the adversarial class and the true class.

$\mathcal{N}$**attack.** $\mathcal{N}$attack (Li et al., 2019) is a derivative-free black-box adversarial attack and it breaks many of the defense methods based on gradient masking. The basic idea is to learn a probability density distribution over a small region centered around the clean input, such that a sample drawn from this distribution is likely to be an adversarial example.

## 2.3 ADVERSARIAL TRAINING

Despite a wide range of defense methods, Athalye et al. (2018) and Li et al. (2019) have broken most previous defense methods (Dhillon et al., 2018; Buckman et al., 2018; Wang & Yu, 2019; Zhang et al., 2019a), and revealed that adversarial training remains one of the best defense method. The basic idea of adversarial training is to solve the min-max optimization problem, as shown in Eq. (4):

$$\min_F \max_{x': \|x'-x\|_\infty < \epsilon} \mathcal{L}\left(F; x', y\right). \tag{4}$$

Here we introduce two currently state-of-the-art adversarial training frameworks.

**PGD adversarial training.** PGD Adversarial Training (PGDAT) (Madry et al., 2018) leverages the PGD attack to generate adversarial examples, and trains only with the adversarial examples. The objective function is formalized as follows:

$$\mathcal{L}_{\text{PGD}}(F; x, y) = \mathcal{L}(F; x'_{\text{PGD}}, y), \tag{5}$$

where $x'_{\text{PGD}}$ is obtained via the PGD attack on the cross entropy $\mathcal{L}(F; x, y)$.

**TRADES.** Zhang et al. (2019a) propose TRADES to specifically maximize the trade-off of adversarial training between adversarial robustness and standard accuracy by optimizing the following regularized surrogate loss:

$$\mathcal{L}_{\text{TRADES}}(F; x, y) = \mathcal{L}(F; x, y) + \lambda D_{\text{KL}}\left(p_F(\cdot|x) \,\|\, p_F\left(\cdot|x'_{\text{PGD}}[x]\right)\right), \tag{6}$$

where $x'_{\text{PGD}}[x]$ is obtained via the PGD attack on the KL-divergence $D_{\text{KL}}\left(p_F(\cdot|x) \,\|\, p_F\left(\cdot|x'\right)\right)$, and $\lambda$ is a hyper-parameter to control the trade-off between adversarial robustness and standard accuracy.

## 3 ROBUST LOCAL FEATURES FOR ADVERSARIAL TRAINING

Unlike adversarially trained models, normally trained models are more biased towards the local features but vulnerable to adversarial examples (Geirhos et al., 2019). It indicates that, in contrast to global structural features, local features seems be more well-generalized but less robust against adversarial perturbation. Based on the basic observation, in this work, we focus on the learning of robust local features on adversarial training, and propose a novel form of adversarial training called RLFAT that learns the robust local features and transfers the robust local features into the training of normal adversarial examples. In this way, our adversarially trained models not only yield strong robustness against adversarial examples but also show great generalization on unseen testing data.

### 3.1 ROBUST LOCAL FEATURE LEARNING

It's known that adversarial training tends to capture global structure features so as to increase invariance against adversarial perturbations (Zhang & Zhu, 2019; Ilyas et al., 2019). To advocate for

the learning of robust local features during adversarial training, we propose a simple and straight-forward image transformation called Random Block Shuffle (RBS) to break up the global structure features of the images, at the same time retaining the local features. Specifically, for an input image, we randomly split the target image into $k$ blocks horizontally and randomly shuffle the blocks, and then we perform the same split-shuffle operation vertically on the resulting image. As illustrated in Figure 1, RBS transformation can destroy the global structure features of the images to some extent and retain the local features of the images.

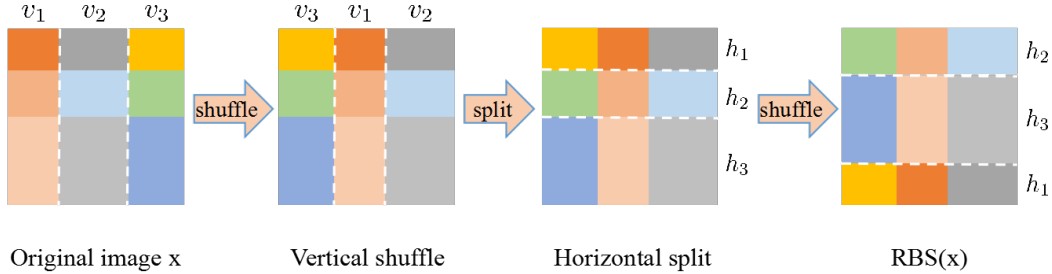

Figure 1: Illustration of the RBS transformation for $k = 3$. For a better understanding on the RBS transformation, we paint the split image blocks with different colors.

Then we apply the RBS transformation on adversarial training. Different from normal adversarial training, we use the RBS-transformed adversarial examples rather than normal adversarial examples as the adversarial information to encourage the models to learn robust local features. Note that we only use the RBS transformation as a tool to learn the robust local features during adversarial training and will not use RBS transformation in the inference phase. we refer to the form of adversarial training as *RBS Adversarial Training* (RBSAT).

To demonstrate the generality of our argument, we consider two currently state-of-the-art adversarial training frameworks, PGD Adversarial Training (PGDAT) (Madry et al., 2018) and TRADES (Zhang et al., 2019a), to demonstrate the effectiveness of the robust local features.

We use the following loss function as the alternative to the objective function of PGDAT:
$$\mathcal{L}_{\text{PGDAT}}^{\text{RLFL}}(F; x, y) = \mathcal{L}(F; \text{RBS}(x'_{\text{PGD}}), y) \,, \tag{7}$$
where $\text{RBS}(\cdot)$ denotes the RBS transformation; $x'_{\text{PGD}}$ is obtained via the PGD attack on the cross entropy $\mathcal{L}(F; x, y)$.

Similarly, we use the following loss function as the alternative to the objective function of TRADES:
$$\mathcal{L}_{\text{TRADES}}^{\text{RLFL}}(F; x, y) = \mathcal{L}(F; x, y) + \lambda D_{\text{KL}}\left[\, p_F(\cdot|x) \,\|\, p_F\left(\cdot|\text{RBS}\left(x'_{\text{PGD}}[x]\right)\right) \,\right] \,, \tag{8}$$

where $x'_{\text{PGD}}[x]$ is obtained via the PGD attack on the KL-divergence $D_{\text{KL}}\left(\, p_F(\cdot|x) \,\|\, p_F\left(\cdot|x'\right) \,\right)$.

### 3.2 ROBUST LOCAL FEATURE TRANSFER

Since the type of input images in the training phase and the inference phase is different (RBS transformed images for training, versus original images for inference), we consider to transfer the knowledge of the robust local features learned by RBSAT to the normal adversarial examples. Specifically, we present a knowledge transfer scheme, called *Robust Local Feature Transfer (RLFT)*. The goal of RLFT is to learn the representation that minimizes the feature shift between the normal adversarial examples and the RBS-transformed adversarial examples.

In particular, we apply RLFT on the logit layer for high-level feature alignment. Formally, the objective functions of robust local feature transfer for PGDAT and TRADES are formalized as follows, respectively:
$$\mathcal{L}_{\text{PGDAT}}^{\text{RLFT}}(F; x, y) = \|f(\text{RBS}(x'_{\text{PGD}})) - f(x'_{\text{PGD}})\|_2^2 \,,$$
$$\tag{9}$$
$$\mathcal{L}_{\text{TRADES}}^{\text{RLFT}}(F; x, y) = \|f(\text{RBS}(x'_{\text{PGD}}[x])) - f(x'_{\text{PGD}}[x])\|_2^2 \,,$$
where $f(\cdot)$ denotes the mapping of the logit layer, and $\|\cdot\|_2^2$ denotes the squared Euclidean norm.

### 3.3 Overall Objective Function

Since the quality of robust local feature transfer depends on the quality of the robust local features learned by RBSAT, we integrate RBSAT and RLFT into an end-to-end training framework, which we refer to as *RLFAT* (Robust Local Features for Adversarial Training). The general training process of RLFAT is summarized in Algorithm 1. Note that the computational cost of RBS transformation (line 7) is negligible in the total computational cost.

---

**Algorithm 1** Robust Local Features for Adversarial Training (RLFAT).

---

1: Randomly initialize network $F(x)$;
2: Number of iterations $t \leftarrow 0$;
3: **repeat**
4:    $t \leftarrow t + 1$;
5:    Read a minibatch of data $\{x_1, ..., x_m\}$ from the training set;
6:    Generate the normal adversarial examples $\{x_1^{adv}, ..., x_m^{adv}\}$
7:    Obtain the RBS-transformed adversarial examples $\{\text{RBS}(x_1^{adv}), ..., \text{RBS}(x_m^{adv})\}$ ;
8:    Calculate the overall loss following Eq. (10).
9:    Update the parameters of network $F$ through back propagation;
10: **until** the training converges.

---

We implement RLFAT in two currently state-of-the-art adversarial training frameworks, PGDAT and TRADES, and have new objective functions to learn the robust and well-generalized feature representations, which we call $\text{RLFAT}_\text{P}$ and $\text{RLFAT}_\text{T}$:

$$\mathcal{L}_{\text{RLFAT}_\text{P}}(F; x, y) = \mathcal{L}_{\text{PGDAT}}^{\text{RLFL}}(F; x, y) + \eta \mathcal{L}_{\text{PGDAT}}^{\text{RLFT}}(F; x, y),$$

$$\mathcal{L}_{\text{RLFAT}_\text{T}}(F; x, y) = \mathcal{L}_{\text{TRADES}}^{\text{RLFL}}(F; x, y) + \eta \mathcal{L}_{\text{TRADES}}^{\text{RLFT}}(F; x, y),$$

(10)

where $\eta$ is a hyper-parameter to balance the two terms.

## 4 EXPERIMENTS

In this section, to validate the effectiveness of RLFAT, we empirically evaluate our two implementations, denoted as $\text{RLFAT}_\text{P}$ and $\text{RLFAT}_\text{T}$, and show that our models make significant improvement on both robust accuracy and standard accuracy on standard benchmark datasets, which provides strong support for our main hypothesis. Codes are available online[1].

### 4.1 Experimental setup

**Baselines.** Since most previous defense methods provide few benefit in true adversarially robustness (Athalye et al., 2018; Li et al., 2019), we compare the proposed methods with state-of-the-art adversarial training defenses, PGD Adversarial Training (PGDAT) (Madry et al., 2018) and TRADES (Zhang et al., 2019a).

**Adversarial setting.** We consider two attack settings with the bounded $\ell_\infty$ norm: the white-box attack setting and the black-box attack setting. For the white-box attack setting, we consider existing strongest white-box attacks: Projected Gradient Descent (PGD) (Madry et al., 2018) and Carlini-Wagner attack (CW) (Carlini & Wagner, 2017b). For the black-box attack setting, we perform the powerful black-box attack, $\mathcal{N}$attack (Li et al., 2019), on a sample of 1,500 test inputs as it is time-consuming.

**Datasets.** We compare the proposed methods with the baselines on widely used benchmark datasets, namely CIFAR-10 and CIFAR-100 (Krizhevsky & Hinton, 2009). Since adversarially robust generalization becomes increasingly hard for high dimensional data and little training data (Schmidt et al., 2018), we also consider one challenging dataset: STL-10 (Coates et al.), which contains $5,000$ training images, with $96 \times 96$ pixels per image.

---

[1] https://github.com/JHL-HUST/RLFAT

**Neural networks.** For STL-10, the architecture we use is a wide ResNet 40-2 (Zagoruyko & Komodakis, 2016). For CIFAR-10 and CIFAR-100, we use a wide ResNet w32-10. For all datasets, we scale the input images to the range of $[0, 1]$.

**Hyper-parameters.** To avoid posting much concentrate on optimizing the hyper-parameters, for all datasets, we set the hyper-parameter $\lambda$ in TRADES as 6, set the hyper-parameter $\eta$ in RLFAT$_P$ as 0.5, and set the hyper-parameter $\eta$ in RLFAT$_T$ as 1. For the training jobs of all our models, we set the hyper-parameters $k$ of the RBS transformation as 2. More details about the hyper-parameters are provided in Appendix A.

## 4.2 Evaluation results

We first validate our main hypothesis: for adversarial training, is it possible to learn the robust local features that have better adversarially robust generalization and better standard generalization?

In Table 1, we compare the accuracy of RLFAT$_P$ and RLFAT$_T$ with the competing baselines on three standard datasets. The proposed methods lead to consistent and significant improvements on adversarial robustness as well as standard accuracy over the baseline models on all datasets. With the robust local features, RLFAT$_T$ achieves better adversarially robust generalization and better standard generalization than TRADES. RLFAT$_P$ also works similarly, showing a significant improvement on the robustness against all attacks and standard accuracy than PGDAT.

Table 1: The classification accuracy (%) of defense methods under white-box and black-box attacks on STL-10, CIFAR-10 and CIFAR-100.

(a) **STL-10.** The magnitude of perturbation is 0.03 in $\ell_\infty$ norm.

| Defense | No attack | PGD | CW | $\mathcal{N}$attack |
|---|---|---|---|---|
| PGDAT | 67.05 | 30.00 | 31.97 | 34.80 |
| TRADES | 65.24 | 38.99 | 38.35 | 42.07 |
| RLFAT$_P$ | 71.47 | 38.42 | 38.42 | 44.80 |
| RLFAT$_T$ | **72.38** | **43.36** | **39.31** | **48.13** |

(b) **CIFAR-10.** The magnitude of perturbation is 0.03 in $\ell_\infty$ norm.

| Defense | No attack | PGD | CW | $\mathcal{N}$attack |
|---|---|---|---|---|
| PGDAT | 82.96 | 46.19 | 46.41 | 46.67 |
| TRADES | 80.35 | 50.95 | 49.80 | 52.47 |
| RLFAT$_P$ | **84.77** | 53.97 | **52.40** | **54.60** |
| RLFAT$_T$ | 82.72 | **58.75** | 51.94 | **54.60** |

(c) **CIFAR-100.** The magnitude of perturbation is 0.03 in $\ell_\infty$ norm.

| Defense | No attack | PGD | CW | $\mathcal{N}$attack |
|---|---|---|---|---|
| PGDAT | 55.86 | 23.32 | 22.87 | 22.47 |
| TRADES | 52.13 | 27.26 | 24.66 | 25.13 |
| RLFAT$_P$ | 56.70 | **31.99** | **29.04** | **32.53** |
| RLFAT$_T$ | **58.96** | 31.63 | 27.54 | 30.86 |

The results demonstrate that, the robust local features can significantly improve both the adversarially robust generalization and the standard generalization over the state-of-the-art adversarial training frameworks, and strongly support our hypothesis. That is, for adversarial training, it is possible to learn the robust local features, which have better robust and standard generalization.

## 4.3 Loss Sensitivity under Distribution Shift

**Motivation.** Ding et al. (2019) and Zhang et al. (2019b) found that the effectiveness of adversarial training is highly sensitive to the "semantic-loss" shift of the test data distribution, such as gamma mapping. To further investigate the performance of the proposed methods, we quantify the smoothness of the models under the distribution shifts of brightness perturbation and gamma mapping.

**Loss sensitivity on brightness perturbation.** To quantify the smoothness of models on the shift of the brightness perturbation, we propose to estimate the Lipschitz continuity constant $\ell_{\mathcal{F}}$ by using the gradients of the loss function with respect to the brightness perturbation of the testing data. We adjust the brightness factor of images in the HSV (hue, saturation, value) color space, which we refer to as $x^b = \mathcal{V}(x, \alpha)$, where $\alpha$ denotes the magnitude of the brightness adjustment. The lower the value of $\ell_{\mathcal{F}}^b(\alpha)$ is, the smoother the loss function of the model is:

$$\ell_{\mathcal{F}}^b(\alpha) = \frac{1}{m} \sum_{i=1}^{m} \|\nabla_x \mathcal{L}(F; \mathcal{V}(x_i, \alpha), y_{true})\|_2 \tag{11}$$

**Loss sensitivity on gamma mapping.** Gamma mapping (Szeliski, 2011) is a nonlinear element-wise operation used to adjust the exposure of images by applying $\tilde{x}^{(\gamma)} = x^\gamma$ on the original image $x$. Similarly, we approximate the loss sensitivity under gamma mapping, by using the gradients of the loss function with respect to the gamma mapping of the testing data. A smaller value indicates a smoother loss function.

$$\ell_{\mathcal{F}}^g(\gamma) = \frac{1}{m} \sum_{i=1}^{m} \|\nabla_x \mathcal{L}(F; x_i^\gamma, y_{true})\|_2 \tag{12}$$

**Sensitivity analysis.** The results for the loss sensitivity of the adversarially trained models under brightness perturbation are reported in Table 2a, where we adopt various magnitude of brightness adjustment on each testing data. In Table 2b, we report the loss sensitivity of adversarially trained models under various gamma mappings. We observe that $RLFAT_T$ provides the smoothest model under the distribution shifts on all the three datasets. The results suggest that, as compared to PGDAT and TRADES, both $RLFAT_P$ and $RLFAT_T$ show lower gradients of the models on different data distributions, which we can directly attribute to the robust local features.

Table 2: The loss sensitivity of defense methods under different testing data distributions.

(a) Loss sensitivity on brightness perturbation for the adversarially trained models.

| Dataset | $\ell_{\mathcal{F}}^b(-0.15)$ / $\ell_{\mathcal{F}}^b(0.15)$ | | | |
| --- | --- | --- | --- | --- |
| | PGDAT | TRADES | $RLFAT_P$ | $RLFAT_T$ |
| STL-10 | 0.85 / 0.82 | 0.46 / 0.48 | 0.32 / 0.34 | **0.22 / 0.23** |
| CIFAR-10 | 1.41 / 1.36 | 0.88 / 0.90 | 0.70 / 0.71 | **0.54 / 0.56** |
| CIFAR-100 | 3.93 / 3.66 | 2.31 / 2.12 | 1.25 / 1.31 | **1.00 / 0.98** |

(b) Loss sensitivity on gamma mapping for the adversarially trained models.

| Dataset | $\ell_{\mathcal{F}}^g(0.8)$ / $\ell_{\mathcal{F}}^g(1.2)$ | | | |
| --- | --- | --- | --- | --- |
| | PGDAT | TRADES | $RLFAT_P$ | $RLFAT_T$ |
| STL-10 | 0.77 / 0.79 | 0.44 / 0.42 | 0.30 / 0.29 | **0.21 / 0.19** |
| CIFAR-10 | 1.27 / 1.20 | 0.84 / 0.76 | 0.69 / 0.62 | **0.54 / 0.48** |
| CIFAR-100 | 2.82 / 2.80 | 1.78 / 1.76 | 1.09 / 1.01 | **0.95 / 0.88** |

## 4.4 ABLATION STUDIES

To further gain insights on the performance obtained by the robust local features, we perform ablation studies to dissect the impact of various components (robust local feature learning and robust local feature transfer). As shown in Figure 2, we conduct additional experiments for the ablation studies of $RLFAT_P$ and $RLFAT_T$ on STL-10, CIFAR-10 and CIFAR-100, where we report the standard accuracy over the clean data and the *average* robust accuracy over all the attacks for each model.

**Does robust local feature learning help?** We first analyze that as compared to adversarial training on normal adversarial examples, whether adversarial training on RBS-transformed adversarial examples produces better generalization and more robust features. As shown in Figure 2, we observe that Robust Local Features Learning (RLFL) exhibits stable improvements on both standard accuracy and robust accuracy for $RLFAT_P$ and $RLFAT_T$, providing strong support for our hypothesis.

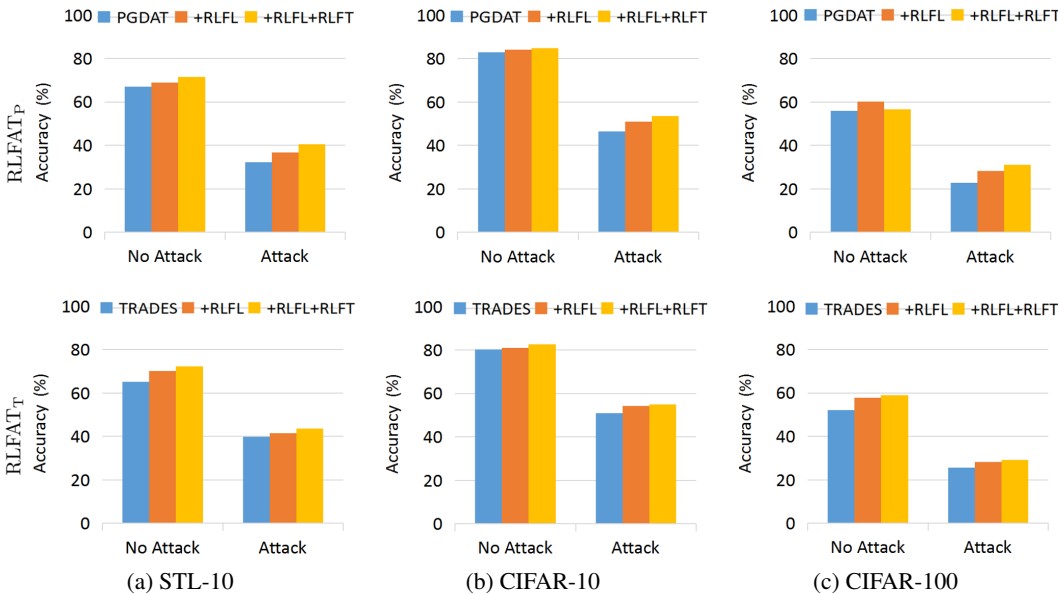

Figure 2: Ablation studies for RLFAT$_P$ and RLFAT$_T$ to investigate the impact of Robust Local Feature Learning (RLFL) and Robust Local Feature Transfer (RLFT).

**Does robust local feature transfer help?** We further add Robust Local Feature Transfer (RLFT), the second term in Eq. (10), to get the overall loss of RLFAT. The robust accuracy further increases on all datasets for RLFAT$_P$ and RLFAT$_T$. The standard accuracy further increases also, except for RLFAT$_P$ on CIFAR-100, but it is still clearly higher than the baseline model PGDAT. It indicates that transferring the robust local features into the training of normal adversarial examples does help promote the standard accuracy and robust accuracy in most cases.

## 4.5 VISUALIZING THE SALIENCE MAPS

We would like to investigate the features of the input images that the models are mostly focused on. Following the work of Zhang & Zhu (2019), we generate the salience maps using *Smooth-Grad* (Smilkov et al., 2017) on STL-10 dataset. The key idea of SmoothGrad is to average the gradients of class activation with respect to noisy copies of an input image. As illustrated in Figure 3, all the adversarially trained models basically capture the global structure features of the object on the images. As compared to PGDAT and TRADES, both RLFAT$_P$ and RLFAT$_T$ capture more local feature information of the object, aligning better with human perception. Note that the images are correctly classified by all these models. For more visualization results, see Appendix B.

## 5 CONCLUSION AND FUTURE WORK

Differs to existing adversarially trained models that are more biased towards the global structure features of the images, in this work, we hypothesize that robust local features can improve the generalization of adversarial training. To validate this hypothesis, we propose a new stream of adversarial training approach called Robust Local Features for Adversarial Training (RLFAT) and implement it in currently state-of-the-art adversarial training frameworks, PGDAT and TRADES. We provide strong empirical support for our hypothesis and show that the proposed methods based on RLFAT not only yield better standard generalization but also promote the adversarially robust generalization. Furthermore, we show that the salience maps of our models on images tend to align better with human perception, uncovering certain unexpected benefit of the robust local features for adversarial training.

Our findings open a new avenue for improving adversarial training, whereas there are still a lot to explore along this avenue. First, is it possible to explicitly disentangle the robust local features from the perspective of feature disentanglement? What is the best way to leverage the robust local

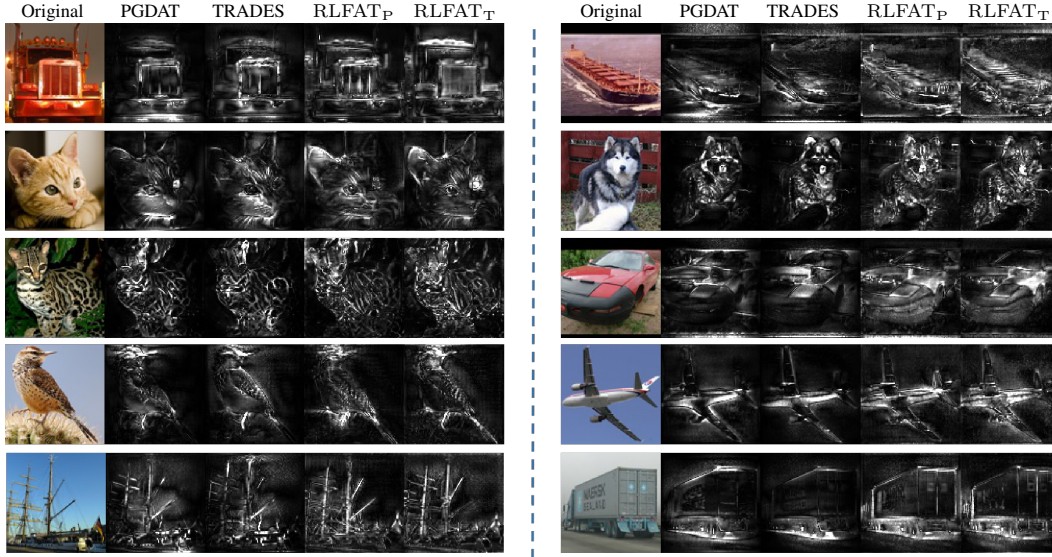

Figure 3: Salience maps of the four models on sampled images. For each group of images, we have the original image, and the salience maps of the four models sequentially.

features? Second, from a methodological standpoint, the discovered relationship may also serve as an inspiration for new adversarial defenses, where not only the robust local features but also the global information is taken into account, as the global information is useful for some tasks. These questions are worth investigation in future work, and we hope that our observations on the benefit of robust local features will inspire more future development.

ACKNOWLEDGMENTS

This work is supported by the Fundamental Research Funds for the Central Universities (2019kfyXKJC021) and Microsoft Research Asia.

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

## A    HYPER-PARAMETER SETTING

Here we show the details of the training hyper-parameters and the attack hyper-parameters for the experiments.

**Training Hyper-parameters.**  For all training jobs, we use the Adam optimizer with a learning rate of 0.001 and a batch size of 32. For CIFAR-10 and CIFAR-100, we run 79,800 steps for training. For STL-10, we run 29,700 steps for training. For STL-10 and CIFAR-100, the adversarial examples are generated with step size 0.0075, 7 iterations, and $\epsilon = 0.03$. For CIFAR-10, the adversarial examples are generated with step size 0.0075, 10 iterations, and $\epsilon = 0.03$.

**Attack Hyper-parameters.**  For the PGD attack, we use the same attack parameters as those of the training process. For the CW attack, we use PGD to minimize its loss function with a high confidence parameter ($k = 50$) following the work of Madry et al. (2018). For the $\mathcal{N}$attack, we set the maximum number of optimization iterations to $T = 200$, $b = 300$ for the sample size, the variance of the isotropic Gaussian $\sigma^2 = 0.01$, and the learning rate $\eta = 0.008$.

## B    MORE FEATURE VISUALIZATION

We provide more salience maps of the adversarially trained models on sampled images in Figure 4.

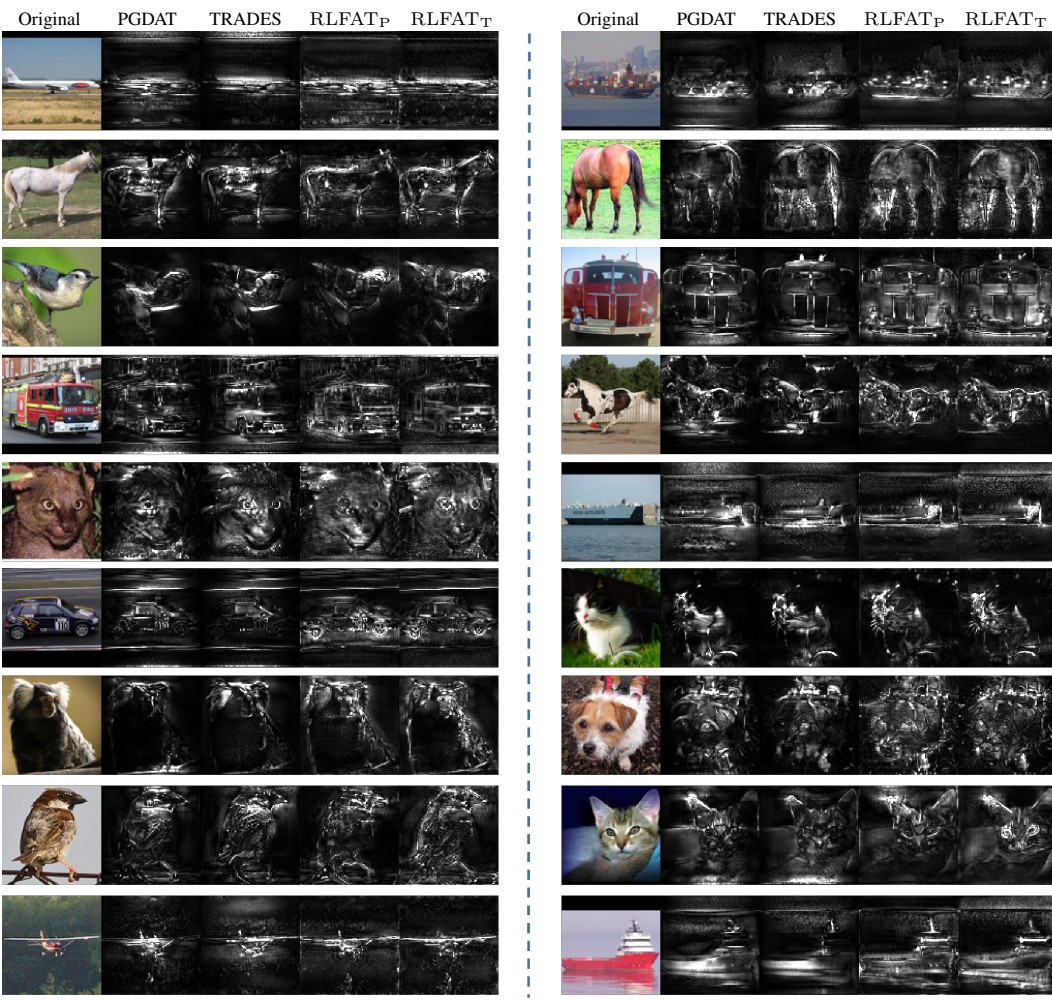

Figure 4: **More Salience maps of the four models.** For each group of images, we have the original image, and the salience maps of the four models sequentially.

