# OpenReview forum: "Robust Local Features for Improving the Generalization of Adversarial Training"
_ICLR.cc/2020/Conference — Accept (Poster)_

### Official Review · AnonReviewer2 · 2019-10-22
**Official Blind Review #2**

**Rating:** 6

**Review:**

In this paper, the authors proposed a new approach to improve the robustness of CNNs against adversarial examples.
The recent studies show that CNNs capture local features, which can be easily affected by the adversarial perturbations.
Thus, in the paper, the authors proposed to train CNNs so that they can capture local features that are robust against the adversarial perturbations.
The difficulty here is that existing adversarial training algorithms tend to bias CNNs to ignore local features and to capture only global features.
To avoid this unfavorable property of the adversarial training, the authors proposed the random block shuffle (RBS) that intensionally destroys the global feature of the images.
The authors demonstrated that combining RBS with the existing adversarial training algorithms can lead to robust CNNs.

I found the paper well-written and the idea is easy to follow.
Especially, the use of RBS seems to be an interesting idea.
As a small downside, the proposed approach looks rather straightforward, and I expect to see any theoretical foundations if possible.

### Updated after author response ###
In summary, the contribution of this study is in twofolds.
1. Proposed an algorithm for learning robust local features.
2. Demonstrated that learning robust local features is effective to improve the robustness of the model.
The possible downside is
3. The proposed approach looks straightforward.
Overall, I like the paper (especially for the reason 2 above), and therefore keep my score.

**Experience Assessment:**

I do not know much about this area.

**Review Assessment: Checking Correctness Of Derivations And Theory:**

N/A

**Review Assessment: Checking Correctness Of Experiments:**

I assessed the sensibility of the experiments.

**Review Assessment: Thoroughness In Paper Reading:**

N/A

---

> ### Author Response · Authors · 2019-11-09
> **RE: Review #2**
>
> We appreciate your time in reviewing our paper and your positive comments. We will try to address your concern as follows.
>
> We thank you for raising the question of formal theoretical foundation. The theoretical foundation of the proposed approach is highly dependent on the theory of the feature disentanglement. The disentanglement of the features from images is a rising direction, such as how to disentangle the robust local features, the features strongly related to label or the features aligning better with human perception from images, and actually there is little disentanglement work [1] in the field of deep learning for adversarial robustness.
>
> While this work shows the general relationship between the generalization of adversarial training and the robust local features, there remains several open questions: how to explicitly disentangle the robust local features? what is the best way to leverage the robust local features for adversarially robust generalization? These are also our future work to make further investigation. We hope this work inspires more work on the theoretical side of the adversarial robustness, and analysis on the disentanglement of robust features.
>
> [1] Adversarial Examples Are Not Bugs, They Are Features. NeurIPS 2019.

---

> > ### Comment · AnonReviewer2 · 2019-11-13
> > **RE: RE: Review #2**
> >
> > I appreciate the authors for the reply to my concern.
> > I understand that the theoretical understanding of adversarial examples is not yet complete.
> >
> > I wonder if this paper can indicate the possible future research directions.
> > For example, some tasks may require global information, e.g. answering "How many apples in the image?" will require a global overview, which may be destroyed by RBS.
> > Thus, training a robust model that can capture both global and local information will be an important direction.
> > I am interested in to see if the current approach can be extended to training such a model.

---

> > > ### Author Response · Authors · 2019-11-14
> > > **RE: RE: RE: Review #2**
> > >
> > > Thank you very much for the quick reply and raising a possible future research direction for us. Following your constructive suggestion, we have performed revision in the conclusion section.
> > >
> > > In this work, in order to explore the relationship between the generalization of adversarial training and the robust local features, we advocate for exploring robust local features by reducing the impact of global information, which is necessary to validate our main hypothesis in this work. However, as you indicated, it does not mean that the global information is useless, as the global information is useful for some tasks, e.g. as you suggested, answering "How many apples in the image".
> > >
> > > We agree that training a robust model that can capture both global and local information will be an important direction. From a methodological standpoint, the discovered relationship may also serve as an inspiration for new adversarial defenses, where not only the robust local features but also the global information is taken into account. One way of introducing the global information would be through adding the term of "adversarial training on original adversarial examples", whereas it is beyond the scope of our main theme (i.e., on the relationship between the generalization of adversarial training and the robust local features) in this work. These questions are indeed worth investigation in future work, and we hope that our observations on the benefit of robust local features will inspire more future development.

---

### Official Review · AnonReviewer1 · 2019-10-23
**Official Blind Review #1**

**Rating:** 3

**Review:**

The work suggests reshuffle images blocks of adversarial examples during adversarial training, in order to improve the generalization performance on benign and adversarial test samples.  The main method is based on the hypothesis in [Zhang et al 2019], [Ilyas et al 2019].  The assumption claims that robust models rely on global structural features, and non-robust models rely on local features. Thus, the work tries to learn local robust features, by cutting and reshuffling the image blocks. Overall the idea is interesting and the paper is well written .

However, there are some concerns about the presentation and the main methodology:
1.	Can the paper give more explanation on the purpose of inserting the feature transfer term in the objective function? What is the difference of the proposed one with directly minimizing the loss on both original PGD image and reshuffled image?
2.	For CIFAR10, TRADEs and PGDAT’s performance in the result is not as good as the ones shown in their original works, which is comparable to the performance of the proposed RLFAT method. More discussions are needed, otherwise the experimental results are not convincing.
3.	More intuitions are needed  on what local and global features are, and why training on the reshuffled images can help learn generalizable robust local features.

Overall the paper is easy to understand, but we suggest that more insight should be given on the success of the proposed method.


**Experience Assessment:**

I have read many papers in this area.

**Review Assessment: Checking Correctness Of Derivations And Theory:**

I carefully checked the derivations and theory.

**Review Assessment: Checking Correctness Of Experiments:**

I carefully checked the experiments.

**Review Assessment: Thoroughness In Paper Reading:**

I read the paper at least twice and used my best judgement in assessing the paper.

---

> ### Author Response · Authors · 2019-11-09
> **RE: Review #1**
>
> Thank you for your insightful comments.  We have performed the corresponding revisions based on your constructive suggestions. And our responses to your concerns are as follows:
>
>
> A1. We have strengthened the motivation of the feature transfer term in the revision, Section 3.2: “Since the type of input images in the training phase and the inference phase is different (RBS transformed images for training, versus original images for inference), we consider to transfer the knowledge of the robust local features learned by RBSAT to the normal adversarial examples.” With this purpose, we insert the feature transfer term in the objective function.
>
> In this work, we advocate for learning the robust local features. The difference of "minimizing the loss on original PGD image" and "the feature transfer term" is the supervised information. Instead of using our proposed feature transfer term, if we directly minimize the loss on the original PGD images, the supervised information is the true label of the images. But as [1] suggests, then the model might explore the global structural features. In contrast, our term uses the robust local feature representation learned from RBS adversarial training as the supervised information for the original PGD images.
>
>
> A2. Sorry for the confusion, and we will try to explain as follows:
>
> Reproducing the results with our released codes might provide more intuitive understanding on the performance. Our code builds on the code framework of Madry’s. To make an apples-to-apples comparison, we re-implement the loss function of TRADES with Tensorflow, and use the same training hyper-parameters for all the methods. The implementation code and all the models are released with the submission.
>
> (a)	The difference of performance for PGDAT:
>
> The iteration step of PGD in Madry's paper is set to 7. For CIFAR10, following the settings in TRADES, the iteration step of PGD to generate adversarial examples is set to 10 in our paper, as we want to use the same parameters for all methods for a fair comparison. In the literature, many researchers also use different steps like 10 [2], or 20 [3] for PGD, as long as they keep the same parameters for all methods. Using 10 indicates that we have a stronger training but at the same time a stronger attack that may reduce the robust accuracy.
>
> (b)	The difference of performance for TRADES:
>
> For TRADES, the authors reported the ‘best’ result of 1/λ=6 rather than the ‘mean’ result of 1/λ=6. Our result (50.95%, 1/λ=6）is slightly higher than the mean result (50.64%) of 1/λ=5 in TRADES’s paper (Table 4). The mean robust accuracy increases slowly by the increasing of 1/λ. Thus, we think the mean result of 1/λ=6 is supposed to be slight higher than 50.64%, which is close to our result of 50.95%.
>
>
> A3. Local features are like the local texture and local edges of the object in an image, while global features are like the global shape of the object and some details are ignored.
>
> Local features seems to be well-generalized on unseen shape variation [4] but less robust against adversarial perturbation. We use RBS transformation to break up the global structure information of the adversarial images, and training on the reshuffled adversarial images can force the model to explore the robust local features. In addition, the random shuffle is a way of augmenting the adversarial examples that can help the generalization.
>
> To provide more intuition, we investigate the features of the input images that the models are focused by the visualization of SmoothGrad. Our model captures more local feature information of the object, aligning better with human perception.
>
> [1]	Interpreting adversarially trained convolutional neural networks. ICML 2019.
> [2]	Are Labels Required for Improving Adversarial Robustness. NeurIPS 2019.
> [3]	Adversarial Training for Free! NeurIPS 2019.
> [4]	ImageNet-trained CNNs are biased towards texture; increasing shape bias improves accuracy and robustness. ICLR 2019.

---

> > ### Author Response · Authors · 2019-11-14
> > **Thanks for your attention.**
> >
> > Dear reviewer #1,  we believe we have addressed your concerns and clarified your points in the rebuttal. Do you have an updated assessment (or concerns) of our paper? Thanks for your consideration.

---

### Official Review · AnonReviewer3 · 2019-11-01
**Official Blind Review #3**

**Rating:** 8

**Review:**

The paper is interested in robustness w.r.t. adversarial exemples.

The authors note that:
* features reflecting the global structure are more robust wrt adversarial perturbations, but generalize less;
* features reflecting the local structure generalize well, but are less robust wrt adversarial perturbations.
In hindsight, these claims are intuitive: adversarial perturbations and unseen shape variations are of the same flavor; one should resist to both or handle both, with the difference that the latter is bound to occur (and should be handled) and the former is undesired (and should be resisted).

The goal thus becomes to define local features that are robust.

The proposed approach is based on
* enforcing the invariance of the intermediate representation through shuffling the blocks of the training images;
* building normal adversarial images x' and deriving the block shuffling RBS such that the x' and RBS(x') are most similar w.r.t. the logit layer
* adding these RBS(x') to the training set;

The idea is nice; the experiments are well conducted and convincing (except for the addition of uniform noise, which is unrealistic; you might consider instead systematic noise mimicking a change of light);
I'd like more details about:
* The computational cost of line 7 in algo (deriving the best RBS).

You might want to discuss the relationship between the proposed approach and the multiple instance setting (as if the image was a bunch of patches).

**Experience Assessment:**

I have read many papers in this area.

**Review Assessment: Checking Correctness Of Derivations And Theory:**

N/A

**Review Assessment: Checking Correctness Of Experiments:**

I assessed the sensibility of the experiments.

**Review Assessment: Thoroughness In Paper Reading:**

I read the paper at least twice and used my best judgement in assessing the paper.

---

> ### Author Response · Authors · 2019-11-09
> **RE: Review #3**
>
> We deeply appreciate the reviewer for the positive, insightful comments and constructive suggestions. We have revised our paper accordingly.
>
> A1. This suggestion is great and realistic, which focuses more on the adversarial robustness in the physical world. In Section 4.3, LOSS SENSITIVITY UNDER DISTRIBUTION SHIFT, we replace the “uniform noise addition” to the “brightness perturbation”. We adopt the brightness perturbation with varied daylight intensity, and present the results in Table 2a in the revision. We observe that, as compared to PGDAT and TRADES, both RLFAT_P and RLFAT_T show lower gradients of the models on different data distributions, which we can directly attribute to the robust local features.
>
>
> A2. We report the running time of 400 iterations of training for RLFL_p with or without line 7, on a single Titan X GPU.
>
> -----------------------------------------------------------------------
> | Dataset	|     RLFL_p     | RLFL_p without line 7  |
> -----------------------------------------------------------------------
> | STL-10	        |	681s	|	680s	                   |
> | CIFAR-10	|	720s	|	719s	                   |
> | CIFAR-100	|	541S	|	540s	                   |
> -----------------------------------------------------------------------
>
> We see that the computational cost is mainly concentrated on the generation of adversarial examples and the optimization of the loss function, and the cost of RBS transformation is negligible in the total computational cost.
>
>
> A3. Thanks for introducing the multiple instance setting [1]. Multiple instance setting usually combines the features of different instances in a bag, where the instances share the label of the original image.
>
> The similarity between the two is that the original image need to be split into several blocks or patches. But intrinsically they are different. First, the goal of multiple instance learning (MIL) is to reduce the effect of noisy labels introduced by data augmentation, whereas the goal of our approach is to explore the robust local features. Second, MIL has a large computational cost, as MIL has multiple input instances in the bag. In contrast, the cost of RBS can be ignored. Third, our objective function differs to MIL that considers the probabilities of the instances, which is more like an ensemble learning. While our approach utilizes the random shuffled blocks as a whole to learn the robust local features.
>
> [1] Multiple Instance Learning Convolutional Neural Networks for Object Recognition. ICPR 2017.

---

### Author Response · Authors · 2019-11-09
**General response**

We deeply appreciate all reviewers for the thorough comments and valuable suggestions, which definitely help the improvement of our paper. We would like to briefly summarize our modification in the updated paper and provide specific response in the individual comment.

Our main modifications are as follows:

- Polish the overall writing; emphasize our motivation and clarify the intuition.

- In Section 4.3, we have reorganized the content and replaced the performance analysis on uniform noise with the performance analysis on brightness perturbation.

- To make the results more convincing, we have supplied the training codes of PGDAT and TRADES in the initial code link (i.e., the files "train_pgdat.py" and "train_trades.py")

- In Section 5 (Conclusion and Future Work), we indicate the possible directions for future research based on our observations and also the suggestion of Reviewer 2 on the benefit of robust local features.

We provide strong empirical support for our hypothesis that robust local features can improve the generalization of adversarial training, and we believe that our observations on the benefit of robust local features in adversarial training are very useful and interesting to the community, as it opens a new avenue for improving adversarial training from the perspective of features.

We hope all our effort can make our paper more comprehensive and address most of your concerns. Thank you very much!


Best regards,

Authors

---

### Decision · Program_Chairs · 2019-12-19

**Decision:**

Accept (Poster)

**Comment:**

Earlier work suggests that adversarial examples exploit local features and that more robust models rely on global features. The authors propose to exploit this insight by performing data augmentation in adversarial training, by cutting and reshuffling image block. They demonstrate the idea empirically and witness interesting gains. I think the technique is an interesting contribution, but empirically and as a tool.